# Epileptogenesis and Tumorigenesis in Glioblastoma: Which Relationship?

**DOI:** 10.3390/medicina58101349

**Published:** 2022-09-26

**Authors:** Jessica Rossi, Francesco Cavallieri, Giuseppe Biagini, Romana Rizzi, Marco Russo, Salvatore Cozzi, Lucia Giaccherini, Anna Pisanello, Franco Valzania

**Affiliations:** 1Clinical and Experimental Medicine PhD Program, University of Modena and Reggio Emilia, 41125 Modena, Italy; 2Neurology Unit, Neuromotor & Rehabilitation Department, Azienda USL-IRCCS of Reggio Emilia, 42123 Reggio Emilia, Italy; 3Department of Biomedical, Metabolic and Neural Sciences, University of Modena and Reggio Emilia, 41125 Modena, Italy; 4Radiation Oncology Unit, Oncological Department and Advanced Technologies, Azienda USL-IRCCS of Reggio Emilia, 42123 Reggio Emilia, Italy

**Keywords:** glioblastoma, tumor-related epilepsy, prognosis, marker, survival, seizures

## Abstract

Epilepsy is reported in 29–52% of patients with glioblastoma (GBM) and has an important role in the natural history of this tumor and patients’ life quality. Although GBM is less epileptogenic than lower-grade gliomas, seizures are usually more difficult to control with common antiseizure medications; drug resistance is found in 20% of cases. Recent studies suggest that seizures at the onset of GBM could be a possible favorable independent prognostic factor in patients. Moreover, a growing body of evidence shows that many molecular mechanisms that influence epileptogenesis often regulate GBM growth and invasiveness, sometimes favoring or counteracting the tumor, respectively. The better-characterized players include glutamate, γ-aminobutyric acid, aquaporin-4, and hypoxia-activated molecules. However, currently available data on the molecular basis of epileptogenesis, tumorigenesis, and their relationship is incomplete or discordant and further research is urgently needed on this topic.

## 1. Introduction

Seizures are one of the most frequent clinical manifestations in patients with brain tumors. Prevalence of tumor-related epilepsy in glioma patients depends on tumor grade, as is estimated at 46–90% in patients with low-grade glioma, and it is lower in grade 3 (42–71%) and glioblastoma (GBM, 29–52%), respectively [1].

GBM is the most frequent malignant tumor of the central nervous system in the adult population and is characterized by an aggressive course [2]. Concurrent temozolomide (TMZ) and radiotherapy (RT) followed by maintenance of TMZ are the current standards of care [3]. Recurrence is inevitable: multiple lesions, which do not show patterns of continuous growth, intraventricular spread, or dissemination, occur in less than 5% of patients [4], however metastases are extremely rare [5].

Despite the development of new radiotherapy techniques, there are a few new drugs that seem to be effective and safe. Nonetheless, the prognosis remains poor [6].

It is common knowledge that epileptogenesis in GBM is partially related to increased intracranial pressure due to mass effect, edema, hypoperfusion, and neoangiogenesis. However, it is also strongly influenced by specific structural and functional changes in the peritumoral cortex, where the creation of an inflammatory microenvironment with increased concentrations of cytokines, chemokines, and growth factors contribute not only to epileptogenesis but also to tumor proliferation and invasiveness [7]. Some evidence suggests that the presence of seizures at the onset of GBM could be a possible independent favorable prognostic factor in patients [8,9]. However, the relationship between seizures and prognosis has been investigated by studies carrying some limitations (e.g., the lack of distinction between IDH-mutant vs. IDH-wildtype tumors) [10]. It remains unclear how the same mechanisms that control tumor behavior may regulate epileptogenesis and how epileptogenesis and tumorigenesis may influence each other.

In this review, we report current knowledge on the biomolecular mechanisms underlying the pathogenesis of tumor-related epilepsy in patients with GBM, focusing on the relationship between epileptogenesis and tumor growth.

## 2. Current Evidence on the Pathogenesis of GBM-Related Epilepsy

When considering other gliomas, the pathogenesis of GBM-related epilepsy is multifactorial and not yet fully understood. Current evidence suggests that the major site of epileptiform discharge development is represented by peritumoral tissue, and an increasingly important role is attributed to the interaction between glioma cells and peritumoral neurons [7,11]. From a structural point of view, it results in neuronal and glial loss, altered blood-brain barrier permeability, and recruitment of astrocytes, microglial cells, and circulating macrophages, as well as increased concentrations of cytokines, such as interleukins (ILs) IL-1β, IL-6, IL-8, tumor necrosis factor (TNF)-α, chemokines, and extracellular matrix remodeling enzymes, including matrix metalloproteinase (MMP)-2 and MMP-9, that promote both tumor proliferation and invasiveness as well as increase seizure susceptibility [10,11]. Moreover, distortion and deafferentation of cortico-subcortical networks are observed even far from the neoplastic lesion, and this may partially explain seizure persistence and recurrence after gross total resection of the tumor [10,11]. Changes in the peritumoral cortex also affect the homeostasis of excitatory and inhibitory neurotransmitters, as well as ionic channel expression. Interestingly, these changes in the peritumoral cortex can occur long before the onset of clinically evident seizures, as recently shown in animal models [12].

### 2.1. Aberrant Glutamate Release by Glioma Cells Promotes Both Epileptogenesis and Tumor Growth

Increased glutamatergic transmission in glioma cells is important in favoring tumor epileptogenesis. In support of this, it is common knowledge that mutation in the isocitrate dehydrogenase (IDH) gene is associated with an increased risk of seizures preoperatively and postoperatively [13,14] and this may be a reason for the higher incidence of epilepsy in astrocytoma and oligodendroglioma, compared to GBM. IDH1 and IDH2 encode for an enzyme of the Krebs cycle which catalyzes the oxidative decarboxylation reaction of isocitrate to α-ketoglutarate [14]. The R132H mutation of the IDH1 gene results in a gain of gene function and promotes the accumulation of D-2-hydroxyglutarate (D-2-HG), which is structurally similar to glutamate [14].

Current evidence supports the view that glioma cells secrete a large amount of the excitatory neurotransmitter glutamate, and this is primarily mediated by the xCT cystine/glutamate antiporter, whose level is increased on the synaptic surface of glioma cells [1,7]. Expression of xCT in glioma cells is correlated with the risk of seizures at onset, but not post-operatively, nor with the risk of developing drug-resistant epilepsy [15]. Furthermore, xCT expression does not appear to have a significant correlation with survival [15]. Therefore, xCT cannot be considered predictive of the clinical course in high-grade glioma. Moreover, peritumoral astrocytes show impaired expression of excitatory amino acid transporter 1 (EAAT1) and EAAT2, which uptake synaptic glutamate, contributing to the accumulation of glutamate in the extracellular space [16].

Furthermore, an increased expression of ionotropic α-amino-3-hydroxy-5-methyl-4-isoxazolepropionic acid (AMPA) and N-methyl-D-aspartate (NMDA) receptor has been observed at the level of tumor cells and in peritumoral astrocytes, which may contribute to the high glutamatergic tone and excitability in the peritumoral cortex [7]. Glutamate may be responsible for different effects on glioma cells and peritumoral neuronal cells, respectively. On the one hand, it promotes glioma growth and invasiveness by interacting with AMPA and NMDA receptors. On the other hand, the interaction with the same receptors on neuronal cells promotes epileptogenesis and excitotoxicity, an effect that leads to neuronal death [7]. In addition, activation of AMPA receptors increases neuronal activity, which in turn promotes: (1) the growth and further progression of glioma via the release of brain-derived neurotrophic factor (BDNF); (2) synaptogenesis to increase the number of synapses between peritumoral neurons and glioma cells, via activation of neuroligin 3 (NLGN3) which, in addition, stimulates the pathways of phosphatidylinositol 3-kinase (PI3K)/AKT/mammalian target of rapamycin (mTOR), Src, Shc-Ras-Raf-Mek-Erk by activating integrin β3, epidermal growth factor receptor (EGFR), fibroblast growth factor receptor (FGFR), vascular endothelial growth factor receptor (VEGFR), and further promoting tumor growth and invasiveness [7].

### 2.2. Low Expression of Glutamine Synthetase (GS) as a Marker of Epilepsy and Overall Survival in GBM

GS is an astrocytic enzyme whose expression is more pronounced in brain regions where glutamatergic synapses are abundant [17]. It catalyzes the conversion of glutamate and ammonia to glutamine. GS is present in normal, reactive, and neoplastic astrocytes and mediates tumor growth proliferation [18]. His deficiency may lead to extracellular glutamate accumulation and seizure generation [17]. A study conducted in 2009 by Rosati et al. showed significantly lower levels of GS in epileptogenic GBM than in non-epileptogenic GBM [17]. Moreover, recent evidence shows that the GS expression pattern strongly correlates with survival among patients with newly diagnosed GBM, and absent/reduced GS expression is significantly associated with longer survival [18].

### 2.3. γ-Aminobutyric Acid (GABA)-Ergic Signaling Promotes Epileptogenesis but Slows Tumor Progression

Another consequence of high glutamate concentrations is the dysregulation of GABAergic activity in the peritumoral neocortex. This is due to the altered expression of chloride transporters on the neuronal cell surface, with an increase in the concentration of Na^+^-K^+^-Cl^−^ cotransporter 1 (NKCC1) and a significant reduction in the concentration of Cl^−^-K^+^ symporter 5 (KCC2) in peritumoral neurons [7]. Therefore, neuronal Cl^−^ concentration rises from 10 to 15 mM. Consequently, the activation of type A GABA (GABA_A_) receptors causes an efflux of Cl^−^ outside the cell, resulting in paradoxical depolarization and functional activation of the neuronal cell [7]. Moreover, the upregulation of several subunits (specifically, α1, α5, β1, β3 subunits) of the GABA_A_ receptor at the level of glioma cells may lead to the impairment of tonic GABAergic inhibition and thus increased central nerve excitability [1]. The effects on glioma cells are not yet well understood, but some authors suggest that GABA may slow tumor growth and progression by acting on GABA_A_ receptors of glioma stem cells (GSCs) [19].

### 2.4. Altered Aquaporin-4 Channel Expression May Favor Epileptogenesis and Glioma Invasiveness

Altered expression of cellular ion channels has been recognized to play a role in the pathogenesis of glioma-related epilepsy. A study conducted by Isoardo et al. (2012) [20] revealed that levels of aquaporin-4 (AQP-4) channel in GBM cells are higher in patients with seizures than in seizure-free individuals, and this seems to be related to post-transcriptional modifications, since, despite similar mRNA levels in the two populations, protein levels are much higher in patients with tumor-related epilepsy [20]. Indeed, the AQP4 channel is responsible for maintaining the homeostasis of water molecules in the intra- and extra-cellular space by regulating the efflux of Cl^−^, K^+^, and glutamate. Increased concentrations of this channel on the surface of tumor cells promote an alteration of water and electrolyte homeostasis, contributing to the development of cytotoxic edema and resulting in a compensatory efflux of K^+^, Cl^−^, and glutamate from the cells. This leads to the depolarization of the neuronal membrane, thereby promoting increased excitability. It is widely accepted that increased expression of AQP4 in GBM cells is also associated with glioma invasion and migration [21], but recent data do not support any prognostic role of this marker in GBM [22].

### 2.5. Decreased Hypoxia-Inducible Factor 1α (HIF-1α)/Signal Transducer and Activator of Transcription 5B (STAT5B) Signaling May Promote Epileptogenesis and Reduce Hypoxia-Induced Tumor Growth

HIF-1α induction in hypoxic conditions provokes a mesenchymal shift in GBM cells, through the activation of other key regulators of transcription, including nuclear factor κ-light-chain-enhancer of activated B cells (NF-κB), CCAAT/enhancer-binding protein β (CEBP-β), and Janus kinase 2 (JAK2) and STAT5B pathway [23]. Therefore, HIF-1α has an important role in tumor growth and carcinogenesis. Moreover, HIF-1α/STAT5B signaling could also be directly linked to tumor epileptogenicity [23]. It is known that STAT5B is a negative regulator of xCT expression, and HIF-1α is mediated by connexin 43 (Cx43), whose augmentation is linked to glioma-related epilepsy. A study by Berendsen et al. (Berendsen et al., 2019) [23] showed that tumor expression of HIF-1α and STAT5B is significantly reduced in patients with GBM and epilepsy, compared to patients without epilepsy [23]. Considering the role of HIF-1α in determining tumor proliferation and invasiveness, as well as the association of high expression of these mesenchymal markers with worse survival of GBM patients [24], the reduced HIF-1α expression could be associated with a less aggressive GBM progression. Moreover, STAT5B protein expression was associated with better survival in the population of patients studied by Berendsen et al. [23].

## 3. Discussion and Conclusions

As previously mentioned, GBM is less epileptogenic than lower-grade gliomas. This may be due to the usually brief duration of the disease, but also to molecular determinants, including the absence of IDH 1-2 mutation. However, when present, seizures in GBM are usually more difficult to control with the common antiseizure medications [25]. This contributes to poorer quality of life in patients [25]. The table in the Appendix A (Table A1) summarizes current knowledge and main challenges concerning the relationship between epileptogenesis and tumorigenesis in glioblastoma. Despite vast research focusing on glioma-related epilepsy, most studies have been performed on low-grade gliomas. Moreover, specific predictive markers of epilepsy, as well as drug-resistance markers are not available for GBM patients. Although current evidence argues against the use of antiseizure medications in patients who have never presented with convulsions, antiseizure prophylaxis is often still used routinely in clinical practice [26]. Given the increased susceptibility of patients with brain tumors to the adverse effects of antiseizure therapy, as well as the potential contribution of antiseizure medications to the impairment of cognitive function in these patients, the choice to start antiseizure therapy should always be made after careful consideration of the risk-benefit ratio [25]. Therefore, the comprehension of epileptogenesis mechanisms and the definition of high-specific molecular markers may help to recognize patients with the highest seizure risk, limiting the use of antiseizure therapy to those cases.

Another important point to clarify is the role of epilepsy in determining the prognosis of GBM. A 2018 meta-analysis pointed out that GBM patients who present epilepsy at diagnosis have a significantly longer overall survival compared to patients who present with other symptoms [9], and this prognostic effect could be independent of treatment with specific anti-epileptic drugs and other confounding factors, including tumor volume at presentation, an earlier detection of the tumor, or IDH1 mutations [8]. However, available data on the molecular mechanisms underlying this prognostic effect is incomplete or discordant, and the prognostic weight of each possible marker of epilepsy remains unclear. Therefore, further studies are needed to clarify the molecular relationship between epileptogenesis and tumorigenesis.

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
