# Peer review of "Epileptogenesis and Tumorigenesis in Glioblastoma: Which Relationship?"

_medicina, 2022, doi:10.3390/medicina58101349_

Round 1

Reviewer 1 Report

Although not exhaustive, the authors focus in their condensed review on an important and understudied aspect of brain tumours. The influence of surgical interventions is very briefly addressed whereas effects of pharmacological interventions that change e.g., the inflammatory (micro)environment of tumours is left out in this short review. Otherwise the article is well written and concise.

Author Response

RESPONSE TO REVIEWER 1

Response to reviewer comments on “Epileptogenesis and tumorigenesis in glioblastoma: which relationship?

Reviewer #1: Although not exhaustive, the authors focus in their condensed review on an important and understudied aspect of brain tumours. The influence of surgical interventions is very briefly addressed whereas effects of pharmacological interventions that change e.g., the inflammatory (micro) environment of tumours is left out in this short review. Otherwise the article is well written and concise.

Response 1: We thank the reviewer for the kind remark. We are very pleased with your appreciation. We have done our best in the attempt to better understand the relationship between epileptogenesis and tumorigenesis in glioblastoma.

Best regards,

Jessica Rossi and Francesco Cavallieri on behalf of the authors

Reviewer 2 Report

Reviewer (Remarks to the Author):

The manuscript entitled “Epileptogenesis and tumorigenesis in glioblastoma: which relationship?” by Rossi J et al. underlines that molecular basis of the epileptogenesis-tumorigenesis relationship are incomplete or discordant and further research is needed on this topic. The premise of the work is very interesting, however in its present version, the manuscript requires several significant areas of improvement before consideration for publication.

1) There are some points either discussed haphazardly or overlooked, need to be discussed properly. I wonder if authors would provide a box (containing some bullet points) addressing some major points/ mechanisms/ challenges and/ or answers of some demanding questions of the discussed area.

2) A graphical abstract needs to prepared and explain the review topic?

3) The text needs careful proof reading.

Author Response

RESPONSE TO REVIEWER 2

Response to reviewer comments on “Epileptogenesis and tumorigenesis in glioblastoma: which relationship?

We thank the reviewer and editor for the thoughtful, comprehensive review of our manuscript.

We have done our best to address each concern and believe that the revisions we have made have improved the manuscript.

 Best regards,

Jessica Rossi and Francesco Cavallieri on behalf of the authors.

Reviewer #2: The manuscript entitled “Epileptogenesis and tumorigenesis in glioblastoma: which relationship?” by Rossi J et al. underlines that molecular basis of the epileptogenesis-tumorigenesis relationship are incomplete or discordant and further research is needed on this topic. The premise of the work is very interesting, however in its present version, the manuscript requires several significant areas of improvement before consideration for publication.

1) There are some points either discussed haphazardly or overlooked, need to be discussed properly. I wonder if authors would provide a box (containing some bullet points) addressing some major points/ mechanisms/ challenges and/ or answers of some demanding questions of the discussed area.

We thank the reviewer for the comments. As suggested, we have provided the required box, with the bullet points listing major issues, hypothesized mechanisms, and challenges, so to address the most significant questions of the discussed topic.

2) A graphical abstract needs to prepared and explain the review topic?

We thank the reviewer for the pointing out this issue. As suggested, we have provided a graphical abstract in order to illustrate the review topic.

3) The text needs careful proof reading.

We thank the reviewer for the pointing out this issue. As suggested, we have carefully proofread the manuscript.